# Chemical fixation creates nanoscale clusters on the cell surface by aggregating membrane proteins

Takehiko Ichikawa [1✉], Dong Wang[1,2], Keisuke Miyazawa[1,3], Kazuki Miyata [1,3], Masanobu Oshima [1,2✉] & Takeshi Fukuma [1,3✉]

Chemical fixations have been thought to preserve the structures of the cells or tissues. However, given that the fixatives create crosslinks or aggregate proteins, there is a possibility that these fixatives create nanoscale artefacts by aggregation of membrane proteins which move around freely to some extent on the cell surface. Despite this, little research has been conducted about this problem, probably because there has been no method for observing cell surface structures at the nanoscale. In this study, we have developed a method to observe cell surfaces stably and with high resolution using atomic force microscopy and a micro-porous silicon nitride membrane. We demonstrate that the size of the protrusions on the cell surface is increased after treatment with three commonly used fixatives and show that these protrusions were created by the aggregation of membrane proteins by fixatives. These results call attention when observing fixed cell surfaces at the nanoscale.

[1] Nano Life Science Institute (WPI-NanoLSI), Kanazawa University, Kanazawa 920-1192, Japan. [2] Division of Genetics, Cancer Research Institute, Kanazawa University, Kanazawa 920-1192, Japan. [3] Faculty of Frontier Engineering, Kanazawa University, Kanazawa 920-1192, Japan. ✉email: tichikawa@staff.kanazawa-u.ac.jp; oshimam@staff.kanazawa-u.ac.jp; fukuma@staff.kanazawa-u.ac.jp

Fixation of cells or tissues is the critical first step for histochemical or cytochemical investigations, and thousands of studies have adopted this method for their research[1–3]. Of commonly used chemical fixatives, aldehyde fixatives such as paraformaldehyde (PFA) and glutaraldehyde (GA) create crosslinking between neighboring proteins, and alcohol fixatives such as methanol (MeOH) fix tissues by dehydration[4–9]. Given this mechanism, it has been speculated that proteins on the cell membrane can move freely to some extent aggregate during the fixation process[10,11]. This has a possibility to cause artefacts by creating pseudo-clusters. Previous studies reported that fixatives can change a part of the cell surface structures at scales of hundreds of nanometers to microns[12–25]. However, no reports investigated surface structures on mammalian cells with a resolution of several nanometers because it is still difficult to observe living cell surfaces at the scale of a few nanometers.

So far, many investigations of the cell surface have been performed using optical microscopes or electron microscopes. Optical microscopes allow us to observe the living cell, but the resolution is more than 200 nm due to the diffraction limit. Even the recently developed super-resolution techniques have a resolution of 10–100 nm, and still not possible to observe a few nanometer structures. On the other side, electron microscopes have a high resolution of less than 1 nm, but these cannot observe the living samples. Atomic force microscopy (AFM) has a high resolution of less than 1 nm on a hard surface, and this is also applicable to the living cell observation[26–30]. However, cell surface molecules have been observed using AFM only on the bacterial cells, which have a relatively hard cell wall, and it has been difficult to image the surface of mammalian cells, probably because its surface is soft and easily fluctuate[30–37]. In this study, to investigate the nanoscale effect by fixatives on the mammalian cell surface, we developed a method to observe living mammalian cell surfaces at the scale of several nanometers using microporous silicon nitride membrane (MPM). We successfully observed sub-10 nm protrusions on the cell surface that are comparable to the single molecule[38–40]. And we report that the size of the protrusions on the cell surface increased after treatment with commonly used chemical fixation methods, and these protrusions were caused by the aggregation of membrane proteins.

## Results

### A method for high-resolution observation of living cell surface using atomic force microscopy

To achieve the nanoscale AFM imaging of the cell surface, we developed a method using commercially available MPM as a sample holder of the transmission electron microscope (https://www.norcada.com/products/porous-membranes). We used a 0.2 µm-thick MPM with 5 µm holes at 10 µm pitch supported by a silicon frame size of 2.6 × 2.6 × 0.2 mm (Fig. 1a–c). We placed the MPM on a 35 mm plastic dish using double-sided tape for the membrane side facing down and cultured human colon cancer DLD-1 cells on the backside of the membrane (Fig. 1f). We confirmed that DLD-1 cell normally grows on MPM (Fig. 1d). Next, we fixed the cell-cultured MPM on the custom-made holder implemented with a perfusion system for the membrane side facing up (Fig. 1f, g) and then observed the cell surface with AFM through a hole in the MPM (Fig. 1h). By supporting the cell membrane around the observation area with the MPM, we expected this method would reduce the surface fluctuation, making it possible to increase imaging quality on the cell surface using AFM. Figure 2a–j show the comparison of AFM images with and without MPM. Figure 2a–e show the results of the DLD-1 surface image without MPM. Large-scale image (Fig. 2a; 2.5 × 2.5 µm), intermediate-scale image (Fig. 2b; 0.5 × 0.5 µm), and the same position of three consecutive small-scale images (Fig. 2c; 100 × 100 nm) acquired every 2 min are shown. Figure 2f–j show the results using MPM. The protrusions marked by arrowheads of the same color in Fig. 2c or h indicate the same protrusions. Because AFM easily creates artefacts, the existence of the same protrusions in the different frames means that these protrusions are not artefacts by AFM but real structures. To quantify these protrusions, we developed an auto-recognition tool that draws a line of the half-maximum height of each protrusion (Supplementary Fig. 1). This tool recognized many protrusions with MPM (Fig. 2i) but mostly failed to recognize them without MPM (Fig. 2d). From a series of 10 frames of images, we confirmed 225 protrusions actually exist with MPM, but only 28 protrusions were present without MPM. Figure 2e or j shows the height profile along the line in c or h. The full-width at half maximum (FWHM) was measured as 19.3 nm without MPM, and 6.3 nm (left) and 5.38 nm (right) with MPM. The sizes of the recognized protrusions with MPM are smaller than the size of the recognized

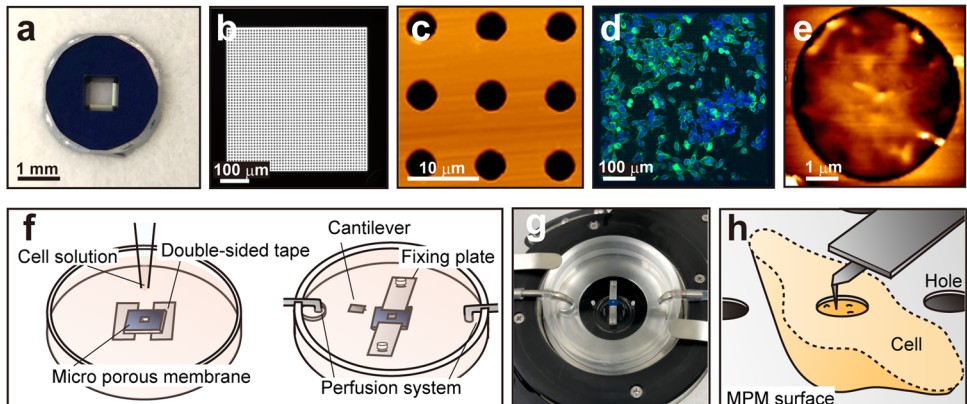

**Fig. 1 Microporous silicon nitride membrane (MPM) and its application to the cell surface observation in AFM imaging. a** Appearance of MPM (NH050D549, Norcada); frame size 2.6 × 2.6 mm, frame thickness 0.2 mm, membrane size 0.5 × 0.5 mm, membrane thickness 200 nm, hole diameter 5 µm, hole pitch 10 µm. **b** The transmitted light image of the membrane. **c** AFM image of the hole of the membrane. **d** Cultured DLD-1 cells on MPM. The cell membrane and nuclei are stained green and blue, respectively. **e** AFM image of the DLD-1 cell surface on the MPM hole. **f** Schematic diagram of the method for culturing and observing the cell surface using MPM. **g** Photo of the area around the sample. **h** Schematic diagram of the AFM observation of the cell surface using MPM.

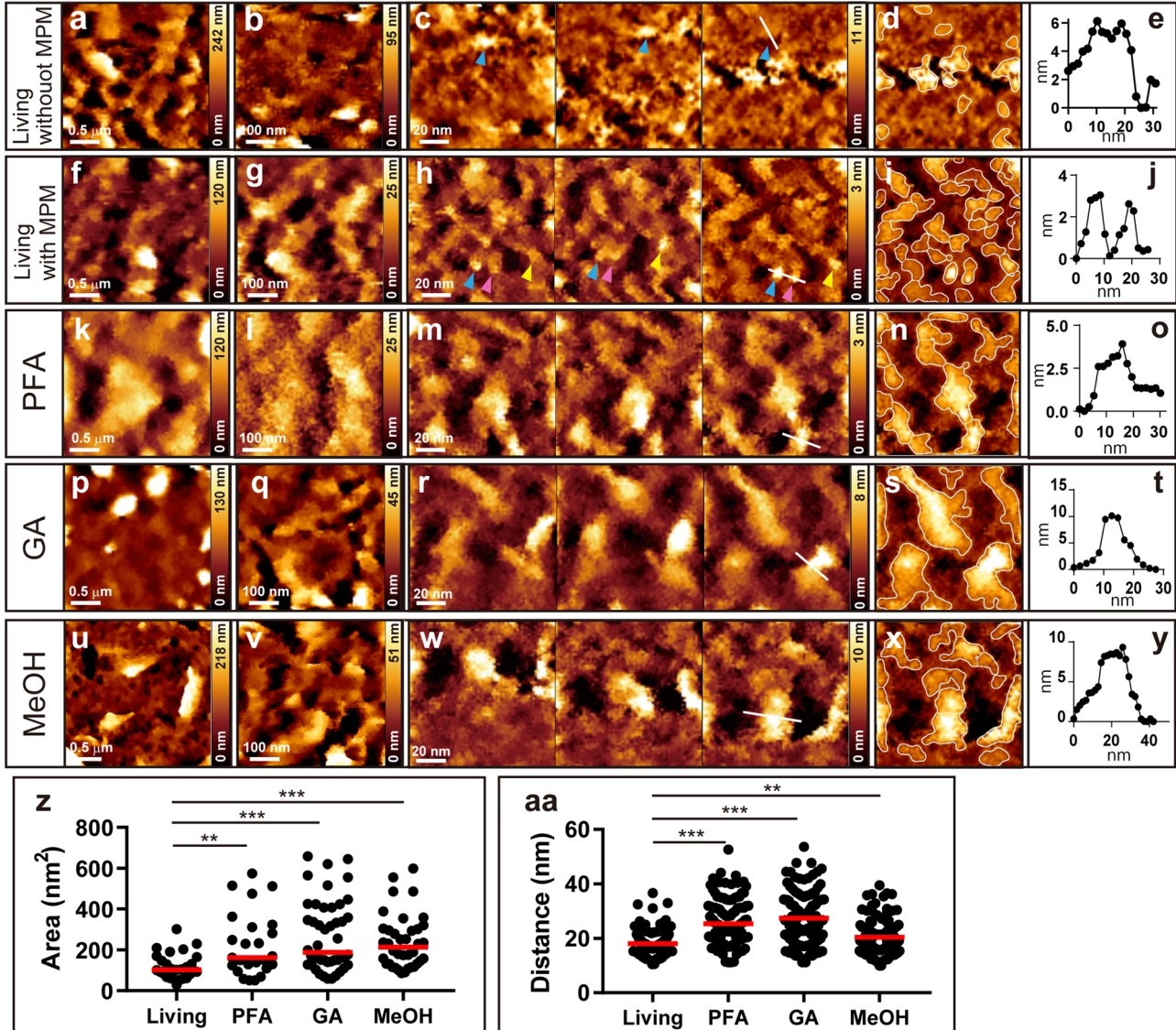

**Fig. 2 AFM images of the DLD-1 cell surface and the effect of the chemical fixation. a** AFM image of the living DLD-1 cell surface without MPM in 2.5 × 2.5 μm scale. **b** 0.5 × 0.5 μm scale image. **c** Three consecutive images of 100 × 100 nm scale at the same position acquired every 2 min. Arrowheads of the same colors indicate the same protrusions. **d** Superimposed image of the third image in **c** with the boundary of the recognized protrusion area (white line) through the auto-recognition tool. **e** Height profile along the line in **c**. **f** AFM image of the living DLD-1 cell surface using MPM in 2.5 × 2.5 μm scale. **g** 0.5 × 0.5 μm scale image. **h** Three consecutive images of 100 × 100 nm scale at the same position acquired every 2 min. Arrowheads of the same colors indicate the same protrusions. **i** Superimposed image of the third image in **h** with the boundary of the recognized protrusion area (white line). **j** Height profile along the line in **h**. **k** AFM image after treatment with 4% PFA using MPM on 2.5 × 2.5 μm scale. **l** 0.5 × 0.5 μm scale image. **m** Three consecutive images of 100 × 100 nm scale. **n** Superimposed image of the third image in **m** with the boundary of the recognized protrusion area (white line). **o** Height profile along the line in **m**. **p** AFM image after treatment with 2% GA using MPM on 2.5 × 2.5 μm scale. **q** 0.5 × 0.5 μm scale image. **r** Three consecutive images of 100 × 100 nm scale. **s** Superimposed image of the third image in **r** with the boundary of the recognized protrusion area (white line). **t** Height profile along the line in **r**. **u** AFM image after treatment with cold 100% MeOH using MPM on 2.5 × 2.5 μm scale. **v** 0.5 × 0.5 μm scale image. **w** Three consecutive images of 100 × 100 nm scale. **x** Superimposed image of the third image in **w** with the boundary of the recognized area (white line). **y** Height profile along the line in **w**. **z** Distributions of the protrusion area on the surface of living or fixed cells using MPM (n = 27 averages per image for living, 25 for PFA, 46 for GA, 38 for MeOH). Red bars indicate median values. Asterisks (** or ***) denote statistical significance (p < 0.01 or p < 0.001, Mann–Whitney U test). **aa** Distributions of the nearest distance between protrusions on the cell surface of living or fixed cells using MPM (n = 129 averages per image for living, 129 for PFA, 114 for GA, 120 for MeOH). Red bars indicate median values. Asterisks (** or ***) denote statistical significance (p < 0.01 or p < 0.001, Mann–Whitney U test). We used BL-AC40TS-C2 cantilevers (Olympus, spring constant ~0.1 N/m).

protrusions without MPM, and the size of the protrusion with MPM is comparable to the size of a single membrane protein[38–40]. Therefore, we are able to stably observe sub-10 nm scale real structures on the surface of living cells by using MPM in AFM imaging.

**Commonly used fixatives create large protrusions on the cell surface.** The results of the AFM observations after treatment with three popular fixatives using MPM are presented in Fig. 2k and later. Figure 2k–o shows the results after treatment with 4% PFA for 30 min at room temperature (RT). Large (2.5 × 2.5 μm),

intermediate ($0.5 \times 0.5$ µm), and small scale ($100 \times 100$ nm) of AFM images are shown in Fig. 2k–m, respectively. Figure 2n is the boundary of the protrusion superimposed image in the third image of Fig. 2m. The size of the protrusions appears to be larger than that of the living cell in Fig. 2i. The height profile measured along the line in Fig. 2m is shown in Fig. 2o, and the FWHM was measured as 11.9 nm, which is approximate twice the size of the living protrusions in Fig. 2j. Figure 2p–t shows the results after treatment with 2% GA for 1 h at RT. The FWHM in Fig. 2t, the heigh profile along the line in Fig. 2r is 9.3 nm. Figure 2u–y depicts the results after treatment with cold 100% MeOH at −20 °C. The FWHM in Fig. 2y, the height profile along the line in Fig. 2w is 16.1 nm. Figure 2z shows the distributions of the protrusion area on the surface of living or fixed cells using MPM ($n = 27$ images for living, 25 for PFA, 46 for GA, 38 for MeOH). Median values of the size of the protrusions in Fig. 2z indicated with red lines are 102.5 (living), 162.1 (PFA), 187.8 (GA), and 213.1 (MeOH) nm$^2$. All fixatives significantly increased the size of the protrusions. Figure 2aa shows the distributions of the nearest distance between protrusions on the cell surface of living or fixed cells ($n = 129$ images for living, 129 for PFA, 114 for GA, 120 for MeOH). The median values of the nearest distances between protrusions were 17.22 (living), 23.37 (PFA), 26.71 (GA), 19.16 (MeOH) nm. To test the possibility of tip contamination[41], we did the same experiments using living and GA fixed cells with a new cantilever (AC40TS-C2). We confirmed that small protrusions were observed in a living cell surface, and only large protrusions were observed after fixation (Supplementary Fig. 2). These results demonstrate that the sizes of the protrusions are significantly increased after treatment with the three fixatives.

Chemical fixation is known to increase the stiffness of the cell surface[18,42–44]. To confirm that chemical fixation was effective in our system, we measured Young's modulus of the cell surface before and after the fixative treatment (Supplementary Fig. 3). We estimated Young's modulus by fitting the Hertz–Sneddon model[45,46] to the approaching force-distance curves at each XY position. The average values of Young's modulus ± SEM of cell surface were as $27.21 \pm 5.43$ (living, $n = 41$), $449 \pm 65.46$ (PFA, $n = 22$), $534.8 \pm 49.7$ (GA, $n = 21$) and $165.3 \pm 11.64$ kPa (MeOH, $n = 38$). Thus, all the fixatives used increased Young's modulus by 6–20 times, which is consistent with previous results[18,20,43,44,47].

**Fluorescence experiments show that the fixatives decrease the nearest distances between irrelevant molecules**. To investigate the possibility that the increased size of the protrusions observed after the fixative treatment was caused by the aggregation of membrane proteins, we used confocal fluorescence microscopy and measured the nearest distances between irrelevant molecules, E-cadherin and the epithelial cell adhesion molecule EpCAM (CD326). These molecules are highly expressed in many cells, and they are reported not to bind each other through direct binding experiments[48]. We labeled E-cadherin and EpCAM using antibodies that bind to extracellular domains and fluorescence-labeled secondary antibodies. Figure 3a–l shows EpCAM, E-cadherin, and overlaid images of living, PFA treated, GA treated, and MeOH treated cells. We cropped the image not to include the boundary and aligned the area per spot in each channel. We measured the distances from the center of the E-cadherin spots to the center of the nearest EpCAM spots and calculated the mean values for each cell and then plotted them in Fig. 3m. The total mean values ± SEM are $0.56 \pm 0.05$ (living, $n = 14$), $0.37 \pm 0.03$ (PFA, $n = 15$), $0.32 \pm 0.02$ (GA, $n = 13$) and $0.39 \pm 0.03$ (MeOH, $n = 14$) µm, respectively. The nearest distances between E-cadherin and EpCAM were significantly decreased after PFA

treatment ($p < 0.01$, $t$-test), GA treatment ($p < 0.001$) and MeOH treatment ($p < 0.05$). We confirmed similar results using different molecules and different cells (Supplementary Figs. 4 and 5) and also confirmed that no significant autofluorescence was observed before and after fixation (Supplementary Fig. 6). Lastly, we observed the movement of membrane molecules during fixation. Figure 3n shows the time series during fixation when GA was added at the 0-time point. After adding GA, the E-cadherin (red) spot moved around for ~10 s and then stopped by aggregating with the EpCAM molecule. Figure 3o shows the time-lapse change of the distance from the indicated E-cadherin to the nearest EpCAM in Fig. 3n. Before adding the fixative, the nearest distance fluctuated by repeating approaching and separating. After adding the fixative, the nearest distance was set to a low value after ~10 s. We also confirmed this aggregation after adding a fixative using a different pair of molecules (Supplementary Fig. 7). Figure 3p shows a model of the behavior of membrane proteins before and after fixation. Before adding fixatives, membrane proteins repeat the approach or move away from each other (left). However, after adding a fixative, it becomes difficult for the proteins to move away from each other once they make contact (right). Membrane proteins rapidly create clusters through the incorporation of the free-moving molecules. As a result, large protrusions are formed on the cell surface, and the mean nearest distance between molecules decreases.

**The positions of the membrane proteins in the fluorescence image are on the large protrusion in the AFM image**. If the large protrusions observed by AFM after fixation result from membrane protein aggregation, then the position of the fluorescence signal of the membrane protein after fixation should mostly correspond to the position of the large protrusion on the cell surface. To confirm this, we investigated whether the fluorescent signal of the membrane protein corresponded to the protrusion of the fixed cell surface in the AFM image. To accurately superimpose the AFM image on the fluorescence images, we developed a method to stain MPMs with fluorescein isothiocyanate and align the edges of the holes. Furthermore, we adopted stimulated emission depletion (STED) microscopy to determine the molecule's position as accurately as possible. Figure 4a, b shows the images of the cell on the MPM with 3-µm holes obtained through AFM and fluorescence microscopy, respectively. Figure 4c shows the overlaid image of Fig. 4a, b. Figure 4d, e shows the cropped AFM images in which the contrast was adjusted and superimposed on the original AFM image and the fluorescence image, respectively. Magnified images of the superimposed area in Fig. 4e are depicted in Fig. 4f. The localization of many E-cadherin signals appears to correspond to the protrusion in the AFM images (Fig. 4f). We obtained similar results when using a different molecule (Supplementary Fig. 8). The size of the protrusions in Fig. 4f appears to be larger than that of Fig. 2r, but this is probably due to the difference in resolution. (Fig. 2r, 1.56 nm per pixel; Fig. 4f, 6.84 nm per pixel). Because there are many membrane proteins other than E-cadherin, large protrusions can be thought to be created by the aggregation of surrounding various kinds of membrane proteins. These results support that the large protrusion in the AFM image after fixation was due to the aggregation of membrane proteins.

**Actin polymerization does not contribute to the creation of large protrusion after fixation**. Our results show that the aggregation of the membrane proteins caused these large protrusions. However, another possible mechanism is that cortical actin polymerization slightly pushed the membrane out at the beginning of the filopodia formation[49,50]. To confirm this

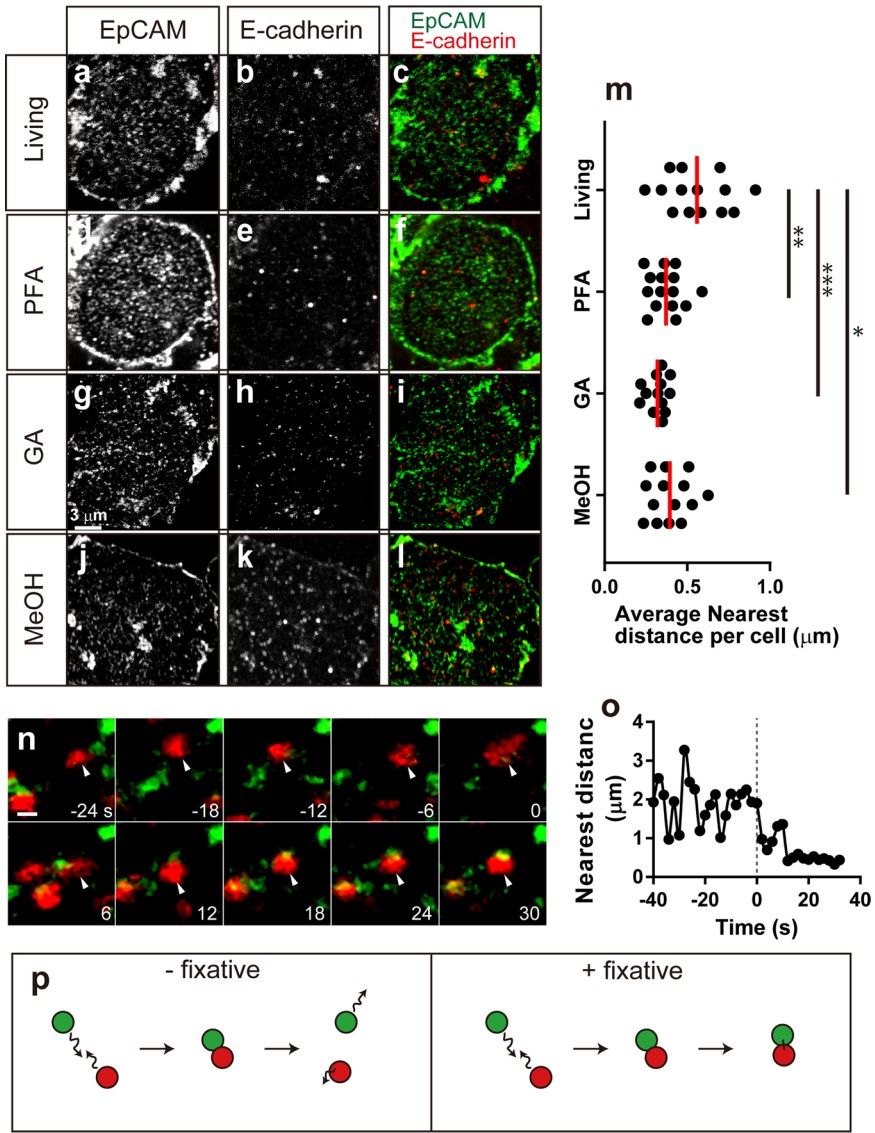

**Fig. 3 Nearest distance of two kinds of membrane proteins and the effect of fixatives. a** EpCAM image of a living cell. **b** E-cadherin image of the same cell as **a**. **c** Superimposed image of EpCAM (**a** green) and E-cadherin (**b** red). **d** EpCAM image after 4% PFA treatment. **e** E-cadherin image of the same cell as **d**. **f** Superimposed image of EpCAM (**d**, green) and E-cadherin (**e** red). **g** EpCAM image after 2% GA treatment. **h** E-cadherin image of the same cell as **g**. **i** Superimposed image of EpCAM (**g** green) and E-cadherin (**h** red). **j** EpCAM image after 100% EtOH treatment. **k** E-cadherin image of the same cell as **j**. **l** Superimposed image of EpCAM (**j**, green) and E-cadherin (**k** red). **m** Distribution of the average nearest distance per cell. $n = 14$ (living), 15 (PFA), 13 (GA) and 14 (MeOH). Red lines indicate mean values. Asterisks (*, ** and ***) denote statistical significance ($p < 0.05$, 0.01 and 0.001, two-sided $t$ test). **n** Time series during fixation. EpCAM (green) and E-cadherin (red) are presented. Arrowheads indicate the same molecule. The scale bar is 1 μm. 2% GA was added at 0 s. **o** Time-lapse change of the distance from the E-cadherin indicated in **n** to the nearest EpCAM. The dotted line indicates the time point of GA addition. **p** Model of the behavior of membrane proteins during fixation. Before adding fixatives, membrane proteins approach and move apart from each other. However, after adding a fixative, proteins cannot move apart once they make contact.

possibility, we did the experiments after treatment of the actin-depolymerization drug, Cytochalasin D (Fig. 5). We treated the DLD-1 cell on MPM with 10 μM Cytochalasin D for 15 min and then fixed it with 2% GA. Figure 5c shows that large protrusions whose size is similar to the protrusions without Cytochalasin D were observed. Therefore, the cortical actin polymerization is unlikely to contribute to the creation of the large protrusion after fixation.

## Discussion

In this study, we developed a method in AFM imaging to stably observe the surface of the cell membrane at the scale of a few nanometers and using this method, we demonstrated that the size of the protrusions on the cell surface observed was increased after treatment with three commonly used fixatives, PFA, GA, and MeOH. The fluorescence study showed that the nearest distance between two molecules is significantly decreased after fixation, and we observed that a membrane molecule stops after binding to other molecules during fixation. We also found that the position of membrane molecule observed by super-resolution microscopy mainly corresponds to the position of the large protrusions in AFM image in fixed cell. And these large protrusions are not created by actin polymerization. These results show that aggregation of the membrane proteins causes large protrusions.

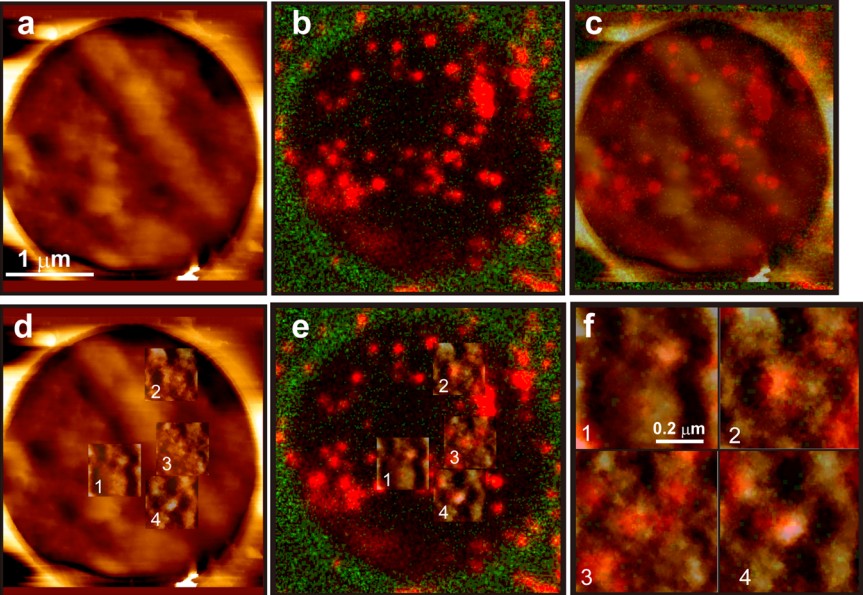

**Fig. 4 Correspondence of AFM and STED image. a** AFM image of DLD-1 cell cultured on 3 μm MPM after fixation using 2% GA. **b** STED image of the same position and scale depicted on **a**. Red spots indicate the localization of E-cadherin. The green color shows the MPM surface. **c** Superimposed image of AFM and fluorescence images. **d** Superimposed image of cropped and contrast adjusted AFM image and original AFM image. **e** Superimposed image of cropped AFM image and STED image. **f** Magnified overlayed image of **e**. Numbers correspond in **e**.

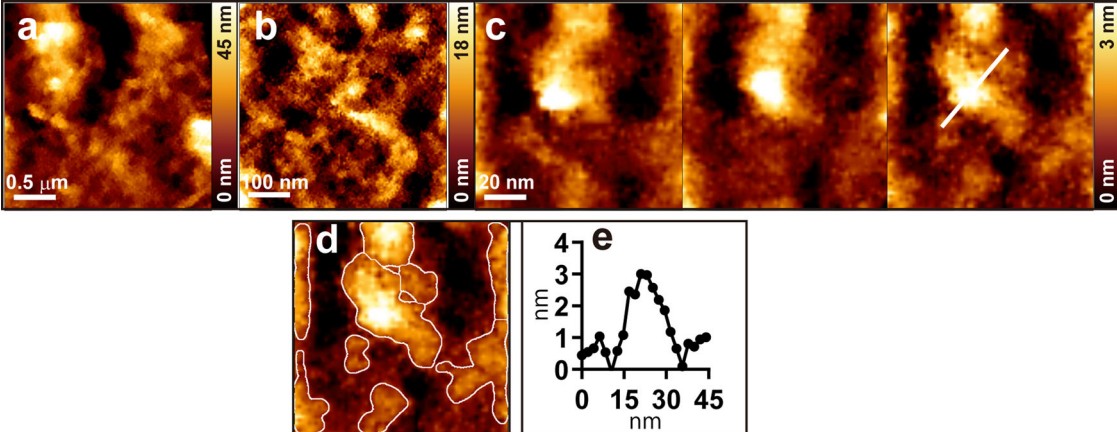

**Fig. 5 AFM measurement after treatment of Cytochalasin D.** We treated the DLD-1 cells on MPM with 10 μM Cytochalasin D for 15 min and then fixed them with 2% GA. **a** 2.5 × 2.5 μm scale image. **b** 0.5 × 0.5 μm scale image. **c** Three consecutive images of 100 × 100 nm. **d** Superimposed image of the third image in **c** with the boundary of the recognized protrusion area (white line). **e** Height profile along the line in **c**. FWHM is 15.6 nm.

Of the three fixatives used in this study, the effect of MeOH was relatively weak and less significant than that of PFA and GA (Figs. 2 and 3). These results can be explained by the difference in the fixation mechanism (Fig. 6). Aldehyde fixatives, such as PFA and GA, directly create crosslinks between membrane proteins. In contrast, alcohol fixatives such as MeOH just remove water between proteins and precipitate them, so the aggregation produced by MeOH is thought to be looser than that of PFA or GA. We consider that the lower significant effect of MeOH as a fixative in the experiments of the nearest distance reflects this difference in the fixation mechanism.

Cell surface protrusions or structures such as microvilli or microridges have been reported so far[51,52]. However, because these structures are much higher than the protrusions reported here (microvillus ~1 μm, microridge ~300 nm) and look very different, the protrusions reported in this study are different from the previously reported microvilli or microridges.

The protrusions after fixation in Fig. 2m, r, w appear to move. We think this is caused by the drift of the AFM imaging. We firmly fixed the MPM and MPM holder at the stage using the fixing plate (Fig. 1f, g). However, the MPM holder is made of acrylic resin, which is softer than glass or metal, and it swells when it is immersed in liquid. Moreover, the cantilever we used (BL-AC40TS-C2, Olympus) is very soft and easily causes deflection displacement due to the change of the surface stress by laser irradiation, and this can cause image drift. We start measurement after confirming that the displacement does not occur, but there could be a small displacement. Therefore, these can cause a slight drift and are detectable when observed on a small scale. We think the cell surface fluidity stopped after fixation. On the other hand, as for the living cell, we suppose that in addition to imaging drift, diffusion of the molecules also contributes because, in the living sample, the protrusions move in random directions, whereas in the fixed sample, all protrusions move uniformly in the same direction (Fig. 2h, m, r, w).

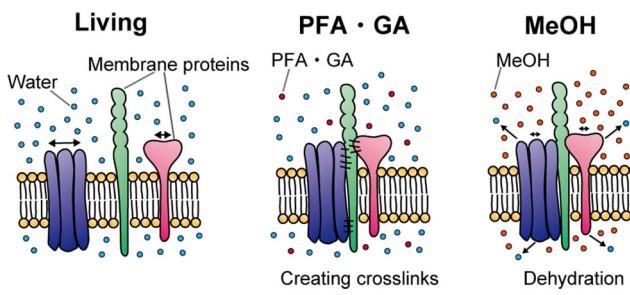

**Fig. 6 Model of the fixation mechanism of membrane proteins using aldehyde or alcohol fixatives.** Aldehyde fixatives (such as PFA and GA) directly create crosslinks between membrane proteins. Thus the nearest distance between membrane proteins is thought to be close. In contrast, alcohol fixatives (such as MeOH) dehydrate and precipitate proteins. Therefore, it can be thought that the binding between membrane proteins is loose, and the nearest distances are not very close. This difference reflects the weak effect of MeOH in Figs. 2 and 3.

Fixatives have been used in thousands of studies, and some have investigated the size of the clusters or nanoscale colocalization of membrane proteins using fixed cells. We suggest that readers are aware that nanoscale clusters and colocalization may include the effect of fixation. Researchers who observe nanoscale clusters also should be careful in interpreting their experimental results when using fixed cells. We recommend that researchers use living cells as much as possible to avoid the effect of fixation when investigating nanoscale clusters of colocalization. Or, if a lipid-only fixation method is developed in the future, it may be possible to achieve a more structure-preserving fixation. Thus, this study should contribute to nanoscale observation research using fixed cells.

## Methods

**Cell sample preparation**. The human colon cancer cell line DLD-1 was supplied by the Cell Resource Center for Biomedical Research, Tohoku University, Japan. The cells were cultured in the Dulbecco's modified Eagle's medium (Gibco) supplemented with 10% fetal bovine serum (Biosera) and 1% penicillin/streptomycin (Fujifilm Wako Pure Chemical Corporation). Microporous silicon nitride membranes (MPM, NH050D549 or NX5100CH3, Norcada) were fixed onto 35 mm-plastic dishes using double-sided tape with the membrane side facing down (Fig. 1f). We placed the 10 µL of the cell suspension solution, which contained $5 \times 10^3$ cells, on the pocket that was on the reverse side of the MPM. The cell solutions were placed in a $CO_2$ incubator and were cultured for 2.5 days. For fluorescence imaging, we seeded $4 \times 10^4$ cells onto a 35-mm glass-bottom dish (Matsunami Glass) and cultured them for 2.5 days in the culture medium.

**Fixation**. The fixatives used in this study were 4% paraformaldehyde (PFA) (Fujifilm Wako Pure Chemical Corporation), 25% glutaraldehyde (GA) (1st grade, Fujifilm Wako Pure Chemical Corporation), and methanol (MeOH) (1st grade, Fujifilm Wako Pure Chemical Corporation). The samples were fixed using 4% PFA for 30 min at RT, 2% GA in PBS for 1 h at RT, or cold 100% MeOH for 20 min at −20 °C.

**Atomic force microscopy (AFM) imaging**. We fixed the cell-cultured MPM onto the MPM holder dish using a perfusion system (custom-made by Nagata Industry Co.) with the membrane facing up. The MPM was bathed in Leibovitz's L-15 medium (Thermo Fisher Scientific) supplemented with 1% penicillin/streptomycin (Fig. 1f). Then, the MPM holder dish was set on the stage of an inverted fluorescence microscope (Eclipse Ti2, Nikon) coupled to a JPK NanoWizard 4 BioAFM (Bruker). The temperature was kept at 37 °C using the heater equipped with this microscope to observe live cells. All AFM imaging was performed using BL-AC40TS-C2 cantilevers (Olympus, spring constant approximately 0.1 N/m). We used the QI settings with following parameters: topography Imaging: $2.5 \times 2.5$ or $0.5 \times 0.5$ µm scale, $64 \times 64$ pixels, Z-length 1 µm, setpoint 0.1 nN, speed 166 µm/s; imaging at the $100 \times 100$ nm scale: $64 \times 64$ pixels, Z-length 50 nm, setpoint 0.1 nN, speed 166 µm/s; for Young's modulus measurement: $100 \times 100$ nm scale, $64 \times 64$ pixels, Z-length 2 µm, setpoint 0.1 nN, speed 166 µm/s; for the experiments of Fig. 4: $3.5 \times 3.5$ µm scale, $512 \times 512$ pixels, Z-length 1 µm, setpoint 0.1 nN, speed 166 µm/s.

**AFM data analysis**. JPK data processing software (ver. 7, Bruker) was used to process the AFM images (plain fitting degree 2, line levelling degree 3, median filter mask width 3, and tolerance 0.5), and they were exported as Tiff images with $512 \times 512$ pixels. These images were imported into a custom-made script written in MATLAB R2020b (MathWorks, available upon request). The area of the protrusions was determined, and the area or the nearest distance between protrusions was measured automatically (Supplementary Fig. 1). The area or the nearest distance distributions were plotted using Prism 7 (GraphPad Software). Young's modulus was calculated using JPK data processing, which employs a Hertz model for a triangular pyramid (angle 17.5°) fitted to the extended curves.

**Fluorescence imaging**. The cells were labeled using primary antibodies, which bind to the extracellular domain of E-cadherin (ab40772, Abcam) or EpCAM (14-9326-82, Thermo Fisher Scientific), EGFR (ab52894, Abcam), ADAM15 (MAB935-SP, R & D Systems), and the following secondary antibodies: STAR RED goat anti-rabbit IgG (STRED-1002, Abberior) or STAR ORANGE goat anti-mouse IgG (STAR ORANGE-1001, Abberior). After culturing the cells on a 35 mm glass-bottom dish, we blocked with Blocking One (Nacalai Tesque) for 30 min at 37 °C and subsequently incubated the cells using the primary antibody solution (1:500 dilution in the culture medium) for 30 min at 37 °C. The cells were washed with warmed PBS and incubated with secondary antibody solution (1:500 dilution in the culture medium) for 30 min at 37 °C. Then they have washed again with warmed PBS, and the culture medium was replaced with Leibovitz's L-15 medium (no phenol red) supplemented with 1% penicillin/streptomycin[53]. In the experiments in Fig. 4, the E-cadherin was labeled using E-cadherin antibody (ab40772, Abcam) and fluorescence-labeled secondary antibody (STRED-1002, Abberior) after AFM observation to avoid the detection of the antibody in AFM imaging.

For confocal imaging, we used the Abberior Expert Line (Abberior Instruments) equipped with an inverted microscope (IX83, Olympus). We used oil immersion 100× lends (NA. 1.3) and illuminated with 561 and 640 nm lasers. We acquired an image with a resolution of $50 \times 50$ nm per pixel. For STED imaging, we used the 2D STED mode. We acquired images with a resolution of $20 \times 20$ nm per pixel. 561 and 640 nm lasers were used for illumination, and 775 nm laser was used for depletion.

For time-lapse imaging, we set the small scan area ($20 \times 17$ µm, $50 \times 50$ nm per pixel) and acquired images every 2 s. A 1/12.5 volume of 25% GA was added during time-lapse imaging by a perfusion system that has been customized for the dish holders (Fig. 1f, g).

**Fluorescence image analysis**. We cropped the image of the cell not to include the cell boundary and aligned the spots density in each channel. The spots were identified using the ImageJ plug-in, Track Mate[54]. We set the size of the spots as 0.5 mm and the threshold at 0.2. We measured the nearest distance between the center of two channel's spots using the position information and a custom script made by MATLAB (R2020b, Simulink). Scatter plots were created using Prism 7 (GraphPad Software). Schematic diagrams in Figs. 1F, 1h, 3p, and 6 were drawn using Illustrator (2022, adobe).

**Statistics and reproducibility**. Data in Fig. 3m, Supplementary Fig. 3, Supplementary Fig. 4, and Supplementary Fig. 5 were analyzed using the two-tailed Student's $t$-test and represented as the mean ± SEM. Data in Fig. 2z and aa are analyzed using Mann–Whitney $U$ test and represented as the median. We examined more than three times for each experiment. We took more than 100 images from five cells for AFM and took more than ten images from ten cells for fluorescence imaging. We did five times for Fig. 3n and four times for Fig. 4.

**Reporting summary**. Further information on research design is available in the Nature Research Reporting Summary linked to this article.

## Data availability

Source data for figures in the paper is available in FigShare: https://doi.org/10.6084/m9.figshare.19609362. The datasets generated during and/or analyzed during the current study are available from the corresponding author on reasonable request.

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

## Acknowledgements

We thank Hiroko Oshima and Mizuho Nakayama for their critical feedback and discussions. We thank Taiki Kitamura for supporting the experiment. We also thank the members of the Fukuma laboratories for revising the manuscript. This work was supported by the World Premier International Research Center Initiative (WPI); JSPS KAKENHI grant number 20H00345, 19K22125 (to T.F.) and 19K06580 (to T.I.); and JST Mirai-Project (No. 18077272, to T.F.).

## Author contributions

T.I. and T.F. designed the experiments. M.O. provided DLD-1 cells. K. Miyazawa and K. Miyata established and maintained the AFM. T.I. and D.W. prepared the samples, and T.I. performed and analyzed the experiments under the supervision of M.O., and T.F. and T.I. wrote the manuscript. All authors read and approved the final manuscript.

## Competing interests

The authors declare no competing interests.
