## [Peer Review File · Communications Biology]

Reviewers' comments:

Reviewer #1 (Remarks to the Author):

The study described in this manuscript aims to investigate the effect of chemical fixation on the cell surface morphology. This is a very important subject and many studies do not mention this step which has important effects on the localization of surface molecules. Several studies have already addressed this subject and are cited in this manuscript (ref 13 and 24). So from this point of view the manuscript here does not bring much novelty on this subject.

The objective of this study is not very well defined. There is a contradiction between the title which refers to the effect of fixation on cluster formation and the objectives described in the introduction and the abstract which describes a new method to observe cells at high resolution. What is really the objective of this study? I think it is important to clarify this.

Here are some remarks and comments that may help to improve the text and the study:

1/ The effects of chemical fixation as studied in this study are interesting . Nevertheless, when we compare the resolution of the images obtained in figure 1a-d and figure 4, we observe a big difference in term of resolution (+/- 10 nm in figure 2 and +/- 100nm in figure 4). Despite the new immobilization method, in figure 4 we do not observe resolution below 100 nm. How can this be explained?

2/ Do you have any idea of the influence of this immobilization method on the behavior of the cells? Could the confinement of the cells in these pores lead to re-localization of the surface receptors? It might be interesting to have controls to validate this.

3/ In the intro on line 36, it is stated that AFM has a resolution of 1nm. This sentence is ambiguous because this value is correct for rigid samples with very sharp tips but not on very soft living cells. The tips used in this study have a theoretical curve radius of 8nm, combined with the indentations, it is probably difficult to explain resolutions below 10 nm.

4/ In figure 2, in the "living" line, we observe movement of the structures which is attributed to the diffusion of the molecules. How to explain after fixation with PFA 4% we still observe movement of the structures (Figure 2h) while in high resolution optical microscopy we do not observe this kind of movement anymore.

Reviewer #2 (Remarks to the Author):

I read the manuscript entitled "Chemical fixation creates nanoscale clusters on the cell surface by aggregating membrane proteins" with great interest. The study tackles the possible artifacts created by the use of fixative solutions when studying the nanoscale architecture of cells and tissues. The authors propose a method to observe the cell surface with high resolution using atomic force microscopy (AFM) and a microporous silicon nitride membrane (MPM). The main conclusion is that chemical fixation leads to the formation of protrusions at the cells surface due to membrane protein aggregation, which can be avoided by using the proposed MPM method. Much of the data provided is interesting and the use of chemical fixation is no doubt of high interest especially due to the intensive use in surperresolution microscopy studies. However, the authors in my view fall short of providing convincing evidence for some of the major conclusions and I believe multiple issues that need to be completely and fully addressed.

I have several major comments:

1) The authors claim that the use of the MPM method reduces surface fluctuation and makes it possible to perform stable high-resolution imaging on the cell surface using AFM. However, no images without the membrane are provided using the same imaging parameters for comparison.

2) How does the mechanical confinement within the MPM affect cells organization at the nanoscale?

3) The Methods section does not mention whether the cells are imaged at 37C. If the temperature

is not close to the physiological one, apoptotic processes can affect cell state and morphology.

4) The use of fixative solutions seems to modify cell surface at the nanoscale. The authors provide a set of images and extract the area and nearest distance between protrusions. It would be useful to mention the number of cells and tips used for this analysis. Also, since tip contamination is a recurring issue when performing high resolution imaging on cells, could the authors compare the results of different tips?

5) The distance between protrusions seems to be highly influenced by how protrusions are defined in the automatic analysis program. It is not clear from the images in Figure 2 that this analysis is correct because there seem to be protrusions of different heights in the maps.

6) The authors claim that the results in Figure 2 demonstrate how fixative solutions generate large protrusions by aggregating mobile proteins. While the size and distribution of the protrusions could be argued and more data could support part of the conclusion, there is no evidence that mobile proteins are forming aggregates. This conclusion is not supported by the data.

7) Following the fluorescence experiments, on Page 6 line 145, the authors claim again that large protrusions are formed on the cell surface through membrane proteins cluster formation. They use confocal microscopy for this quantification, which is limited by the diffraction limit and does not allow the observation of single molecules. Again, the number of cells used for the quantification of the results should be mentioned and possible cell-to-cell differences noted.

8) In the STED-AFM experiments, it is not clear if the cells are fixed or not. Additionally, the authors note that other proteins besides E-cadherin could aggregate as well and conclude that the large protrusions in the AFM image after fixation appear due to membrane protein aggregation. There is no evidence for this conclusion, and it is just a proposed mechanism that would explain the appearance of protrusions.

9) In the Discussion section, actin polymerization pushing the membrane out is mentioned as a possible mechanism of protrusion formation but ruled out because protrusions appear on the entire cellular surface. There is no evidence in the manuscript about the distribution of protrusions on the cell surface. Can the authors rule out the contribution of actin with a depolymerizing drug?

Minor comments:

1) Page 3 line 31: "Optical microscopes are possible to observe..." This should be rephrased.

2) The term "special resolution" is mentioned several times in the introduction. I believe the authors meant to say "lateral resolution"?

3) Page 3 lines 34-35. The sentence implies that 1 nm lateral resolution can be obtained on living cells.

4) The authors use mean values and t-test for the statistical analysis of their data. Did they check for normality? In Figure 2 if the distribution is not normal, the median should be used and a non-parametric test.

5) Page 5 line 104: "Chemical fixation is known to increase the elasticity". Should read decrease the elasticity.

6) Page 12 line 304: the images were acquired every 1 minute. The main text says 2 minutes.

7) The fluorescence experiments require some controls (i.e. no labeling and mock treatment) and separate images showing only E-cadherin and Ep-CAM labeling. More examples could be shown in the Supplementary Information data.

8) Captions should mention the number of experiments, sample preparations, AFM tips.

Authors' reply to comments on "*Chemical fixation creates nanoscale clusters on the cell surface by aggregating membrane proteins*".

Reviewer #1

The study described in this manuscript aims to investigate the effect of chemical fixation on the cell surface morphology. This is a very important subject and many studies do not mention this step which has important effects on the localization of surface molecules. Several studies have already addressed this subject and are cited in this manuscript (ref 13 and 24). So from this point of view the manuscript here does not bring much novelty on this subject.

We thank Reviewer #1 for revising our manuscript. As Reviewer #1 mentioned, the effect of the fixation has been addressed in some papers (ref 13 and 24, etc.). However, to our knowledge, all of these papers observed cellular phenomena at the scale of more than several hundred nanometers, and there is no report observed at the resolution of a few nanometer scales. In this paper, we first provide evidence that fixatives create clusters of several tens nanometers on the cell membrane by direct observation using a new technique of AFM. We believe these results have high novelty because this is the first report describing the creation of nanoscale clusters on the cell membrane by fixatives. Furthermore, these results can influence the research observing the nanoscale clustering or co-localization on cell surfaces after fixation.

The objective of this study is not very well defined. There is a contradiction between the title which refers to the effect of fixation on cluster formation and the objectives described in the introduction and the abstract which describes a new method to observe cells at high resolution. What is really the objective of this study? I think it is important to clarify this.

In this paper, we wanted to report the nanoscale effect of fixation on cluster formation firstly. To achieve this, we developed a new method to observe nanoscale cell surfaces with AFM. We modified the text to clarify this point (Page 3, Line 25).

1/ The effects of chemical fixation as studied in this study are interesting. Nevertheless, when we compare the resolution of the images obtained in figure 1a-d and figure 4, we observe a big difference in term of resolution (+/- 10 nm in figure 2 and +/- 100nm in figure 4). Despite the new immobilization method, in figure 4 we do not observe resolution below 100 nm. How can this be explained?

We appreciate Reviewer #1 for this pointing out. We think there are two reasons. One is that Figure 4 is fixed cells, while Figure 2 f-h (after modification) is living cells. Figure 4 corresponds to Figure 2p-r rather than Figure 2 f-h. We apologize for this lack of information. We modified the caption in Figure 4. Another reason is the difference in the resolution. Figure 4 (3.5 x 3.5 μm scale) was acquired at 512 x 512 pixels (6.84 nm per pixel). Figure 2r (100 x 100 nm scale) was acquired at 64 x 64 pixels (1.56 nm per pixel). Figure 4 needed to include the edge of the 3 μm size of the hole for aligning to the fluorescence image and could not acquire the image at such a high resolution as Figure 2. These are the reasons that the resolution looks different in Figure 2p-r and Figure 4. We have added this explanation in the text (Page 7, Line 23).

2/ Do you have any idea of the influence of this immobilization method on the behavior of the cells? Could the confinement of the cells in these pores lead to re-localization of the surface receptors? It might be interesting to have controls to validate this.

We think MPM itself does not directly affect the movement of surface receptors because there are ECM layers between the cell membrane and MPM surface produced by DLD-1 cell¹, and the thickness of the ECM layer is more than 10 nm². On the other hand, ECM interferes with the diffusion of the membrane molecules³. Because the amount of ECM can be thought to be less in a hole, the diffusion of the membrane molecule may be a little higher in the hole of MPM. This is an interesting topic, as Reviewer #1 mentioned, and can be a future research target.

3/ In the intro on line 36, it is stated that AFM has a resolution of 1nm. This sentence is ambiguous because this value is correct for rigid samples with very sharp tips but not on very soft living cells. The tips used in this study have a theoretical curve radius of 8nm, combined with the indentations, it is probably difficult to explain resolutions below 10 nm.

We agree with this point. We corrected this sentence (Page 3. Line 20).

4/ In figure 2, in the “living” line, we observe movement of the structures which is attributed to the diffusion of the molecules. How to explain after fixation with PFA 4% we still observe movement of the structures (Figure 2h) while in high resolution optical microscopy we do not observe this kind of movement anymore.

As Reviewer #1 mentioned, the movement of the protrusions after fixation was observed in Figure 2m (after modification). We think this is caused by the drift of the AFM imaging. We firmly fixed the MPM and MPM holder at the stage using the fixing plate (Figure 1f and g). However, the MPM holder is made of acrylic resin, which is softer than glass or metal, and it swells when it is immersed in liquid. Moreover, the cantilever we used (BL-AC40TS-C2, Olympus) is very soft and easily cause deflection displacement due to the change of the surface stress by laser irradiation, and this can cause image drift. We start measurement after confirming that the displacement does not occur, but there could be a small displacement. Therefore, these can cause a slight drift and are detectable when observed on a small scale. We think the cell surface fluidity stopped after fixation. As for the living cell, we suppose that in addition to imaging drift, diffusion of the molecules also contributes because, in the living sample, the protrusions move in random directions, whereas in the fixed sample, all protrusions move uniformly in the same direction (Figure 2h, m, r, w). To discuss this, we add a paragraph in the discussion (Page 8. Line 33).

Reviewer #2

I read the manuscript entitled “Chemical fixation creates nanoscale clusters on the cell surface by aggregating membrane proteins” with great interest. The study tackles the possible artifacts created by the use of fixative solutions when studying the nanoscale architecture of cells and tissues. The authors propose a method to observe the cell surface with high resolution using atomic force microscopy (AFM) and a microporous silicon nitride membrane (MPM). The main conclusion is that chemical fixation leads to the formation of protrusions at the cells surface due to membrane protein aggregation, which can be avoided by using the proposed MPM method. Much of the data provided is interesting and the use of chemical fixation is no doubt of high interest especially due to the intensive use in superresolution microscopy studies. However, the authors in my view fall short of providing convincing evidence for some of the major conclusions and I believe multiple issues that need to be completely and fully addressed.

We thank Reviewer #2 for the revision of our manuscripts. We seriously received all suggestions and comments and replied to all point-by-point. We added new seven figures and made major changes in Figure 2, 3 and the relevant sections. With the utmost courtesy, we would like to point out one point. Summary mentioned that MPM could help avoid the creation of the protrusions, but we did not intend to claim this point. Instead, we intended to claim that the MPM can stabilize

AFM measurement of the living cell surface, which helped us find that chemical fixation can create protrusions. We hope that this explanation can clarify our main claims.

1) The authors claim that the use of the MPM method reduces surface fluctuation and makes it possible to perform stable high-resolution imaging on the cell surface using AFM. However, no images without the membrane are provided using the same imaging parameters for comparison.

We appreciate this comment and apologize that we did not provide the images without the MPM. The reason that we did not compare the images with and without MPM is that it is principally difficult to exactly compare the two images because the apical side is observed without using MPM, and the basal side is observed when using MPM. Since DLD-1 has an apico-basal polarity⁴, we had been feared that the comparison between the two images could mean that we are looking at different things. On the other hand, we had noticed that the use of MPM much increases measurement stability. To exhibit this increased stability in AFM observation, we decided to show the results without MPM (Figure 2a-e). Because AFM can easily create artefacts, the protrusion is proven to actually exist only when the same protrusion is confirmed in different frames. From a series of 10 frames of images, we confirmed that 225 protrusions were actually present when we used MPM, and only 28 protrusions were observed when we did not use MPM. Of course, this result may include the difference in observation location of the cell, but in our experiment, the use of MPM enabled us to stably observe the detail change of the cell surface at the scale of a few nanoscales. We added the images without MPM and corrected the text and caption according to this modification (Page 4).

2) How does the mechanical confinement within the MPM affect cells organization at the nanoscale?

We think MPM itself does not confine the membrane molecule in a hole of MPM because there are ECM layers between the cell membrane and MPM, and the thickness is more than 10 nm². On the other hand, ECM interaction to the cell membrane is reported to decrease the diffusion of the membrane molecule³. Since the amount of ECM can be thought to be less in areas of holes than areas of MPM, the diffusion of the membrane molecules may be increased in areas of holes. We think that MPM makes the cell surface in the area of a hole flat and reduces the fluctuation, and these enable stable imaging of the living cell surface in AFM. In any case, we think that MPM does not affect the creation of large protrusions by the fixatives.

3) The Methods section does not mention whether the cells are imaged at 37C. If the temperature is not close to the physiological one, apoptotic processes can affect cell state and morphology.

We apologize for this lack of information. We maintained the temperature at 37 °C using the heater equipped to the stage for the observation of living cells. We added this information in the Methods (Page 10, Line 18).

4) The use of fixative solutions seems to modify cell surface at the nanoscale. The authors provide a set of images and extract the area and nearest distance between protrusions. It would be useful to mention the number of cells and tips used for this analysis. Also, since tip contamination is a recurring issue when performing high resolution imaging on cells, could the authors compare the results of different tips?

We added information about the number of the data and the tip. We also added an additional result using different tips (Supplementary Figure 2). The size of the protrusion of the living DLD-1 cell is about 6.2 nm and 15.3 nm after fixation. These are almost the same results with Figure 2f-j and Figure 2p-t. We added this result in the text (Page 5, Line 23).

5) The distance between protrusions seems to be highly influenced by how protrusions are defined in the automatic analysis program. It is not clear from the images in Figure 2 that this analysis is correct because there seem to be protrusions of different heights in the maps.

The automatic analysis program recognizes each peak of the protrusions and draws the lines at the half-height. Therefore, these heights are different among protrusions. We confirmed that this program recognizes almost the same positions as the line manually recognized (Supplementary Figure 1). To make this clear, we modified this explanation in the text (Page 4, Line 23) and the caption of Supplementary Figure 1.

6) The authors claim that the results in Figure 2 demonstrate how fixative solutions generate large protrusions by aggregating mobile proteins. While the size and distribution of the protrusions could be argued and more data could support part of the conclusion, there is no evidence that mobile proteins are forming aggregates. This conclusion is not supported by the data.

In Page 5, we concluded that fixatives created large protrusions and tried to explain the difference between with and without fixation in Figure 2aa by suggesting a hypothesis that aggregating mobile membrane proteins make large protrusions. Therefore, this is not a conclusion and only a hypothesis at this point. This hypothesis is examined in Figure 3 and 4. Figure 3a-m demonstrate the distance between irrelevant two kinds of molecules decreased after fixation, and this was also confirmed by using different molecules (Supplementary Figure 4) and different cell (Supplementary Figure 5). Figure 3n and o directly show the membrane protein aggregate during fixation, and this was also observed when using other molecules (Supplementary Figure 6). Figure 4 and Supplementary Figure 7 show that the protrusions after fixation mainly correspond to the localization of membrane molecules. These data indicate that the protrusion after fixation was created by the aggregation of membrane proteins. However, since the sentence on Page 5 can be misleading, we moved and merged it to the conclusion (Page 8, Line 9).

7) Following the fluorescence experiments, on Page 6 line 145, the authors claim again that large protrusions are formed on the cell surface through membrane proteins cluster formation. They use confocal microscopy for this quantification, which is limited by the diffraction limit and does not allow the observation of single molecules. Again, the number of cells used for the quantification of the results should be mentioned and possible cell-to-cell differences noted.

We thank Reviewer #2 for this comment. In Figure 3m and o, we measured the distance between the centre of two spots of different channels acquired at the resolution of 50 nm per pixel. Therefore, even if the spot size is more than the diffraction limit (~200 nm), the distances can be lower than this length. And each bright spot may be an aggregate of several molecules, as Reviewer #2 mentioned, so the results in Figure 3m can be thought to be distances among the mixture of single molecules and aggregated molecules.

We added the number of cells in the caption of Figure 3. We also added the new results using different cells (Supplementary Figure 5). When we used the HeLa cell, the nearest distance was also significantly decreased after fixation. We modified the related sentences of the text (Page 6).

8) In the STED-AFM experiments, it is not clear if the cells are fixed or not. Additionally, the authors note that other proteins besides E-cadherin could aggregate as well and conclude that the large protrusions in the AFM image after fixation appear due to membrane protein aggregation. There is no evidence for this conclusion, and it is just a proposed mechanism that would explain the appearance of protrusions.

We apologize for this lack of information about fixation. Figure 4 shows the cell after fixation using 2% GA. We added this information to the text, caption and method. We also added new results using EGFR (Supplementary Figure 7). These results also show that most positions of

the EGFR molecule correspond to the protrusion in the AFM image. We modified the text, and we also changed the expression according to Reviewer #2's comment (Page 7, Line 27).

9) In the Discussion section, actin polymerization pushing the membrane out is mentioned as a possible mechanism of protrusion formation but ruled out because protrusions appear on the entire cellular surface. There is no evidence in the manuscript about the distribution of protrusions on the cell surface. Can the authors rule out the contribution of actin with a depolymerizing drug?

We added the new results of AFM measurement after treatment of actin-depolymerizing drug (Figure 5). We treated the DLD-1 cell on MPM with 10 μ M Cytochalasin D for 15 min and then fixed it with 2% GA. Figure 5c shows the consecutive images of 100 x 100 nm scale. Figure 5e shows the height profile along the line in (c). FWHM is 15.6 nm. We observed only large protrusions on the cell surface after treatment of actin-depolymerizing drug, and also, the size of this protrusion was similar without the actin-depolymerizing drug. Therefore, actin-polymerization is not likely to contribute to the formation of the large protrusion after fixation. We added these results in the text (Page 7, Line 31).

Minor comments:

1) Page 3 line 31: "Optical microscopes are possible to observe..." This should be rephrased.

We rephrased this sentence (Page 3, Line 15).

2) The term "special resolution" is mentioned several times in the introduction. I believe the authors meant to say "lateral resolution"?

We deleted "special" in the manuscript (Page 3, Line 16).

3) Page 3 lines 34-35. The sentence implies that 1 nm lateral resolution can be obtained on living cells.

We modified this sentence (Page 3, Line 20).

4) The authors use mean values and t-test for the statistical analysis of their data. Did they check for normality? In Figure 2 if the distribution is not normal, the median should be used and a non-parametric test.

We appreciate Reviewer #2 for this pointing out. We tested D'Agostino and Pearson normality test on the samples used in Figure 2 and found that some samples were not normally distributed. Therefore, we changed the analytical method to the median and Mann-Whitney U test. We modified the related sentences of the text and caption.

5) Page 5 line 104: "Chemical fixation is known to increase the elasticity". Should read decrease the elasticity.

We corrected this sentence (Page 5, Line 29).

6) Page 12 line 304: the images were acquired every 1 minute. The main text says 2 minutes.

2 minutes is true. We corrected this (Figure 2 caption).

7) The fluorescence experiments require some controls (i.e. no labeling and mock treatment) and separate images showing only E-cadherin and Ep-CAM labeling. More examples could be shown in the Supplementary Information data.

We thank Reviewer #2 for this comment. We added the no labelling image of the living and fixed DLD-1 cell (Supplementary Figure 8). And we changed the representation method of the fluorescence experiments in Figure 3. We showed separate images showing only E-cadherin, Ep-CAM labelling, and overlaid images. We also used the mean values of each cell for plotting the graph. We added new data using different molecules (Supplementary Figure 4), different cells (Supplementary Figure 5), and different molecules for Figure 3n and o (Supplementary Figure 6). We corrected the corresponding sections in the text and caption.

8) Captions should mention the number of experiments, sample preparations, AFM tips.

We apologize for these points. We added the number of samples, sample preparation, and AFM tip information (captions in Figure 2, 3 and Supplementary Figure 3).

1. Klijn C, *et al.* A comprehensive transcriptional portrait of human cancer cell lines. *Nat Biotechnol* **33**, 306-312 (2015).
2. Chen LB, Murray A, Segal RA, Bushnell A, Walsh ML. Studies on intercellular LETS glycoprotein matrices. *Cell* **14**, 377-391 (1978).
3. Kihara T, Ito J, Miyake J. Measurement of biomolecular diffusion in extracellular matrix condensed by fibroblasts using fluorescence correlation spectroscopy. *PLoS One* **8**, e82382 (2013).
4. Fujiwara M, *et al.* Epithelial DLD-1 Cells with Disrupted E-cadherin Gene Retain the Ability to Form Cell Junctions and Apico-basal Polarity. *Cell Struct Funct* **40**, 79-94 (2015).

REVIEWERS' COMMENTS:

Reviewer #1 (Remarks to the Author):

The authors addressed my previous comments.

Reviewer #3 (Remarks to the Author):

The authors addressed my comments and improved the manuscript. I believe this work is now ready for publishing in Communications Biology.